# Demystifying Network Foundation Models

**Sylee (Roman) Beltiukov**[*]
UC Santa Barbara

**Satyandra Guthula**
UC Santa Barbara

**Wenbo Guo**
UC Santa Barbara

**Walter Willinger**
NIKSUN, Inc

**Arpit Gupta**
UC Santa Barbara

## Abstract

This work presents a systematic investigation into the latent knowledge encoded within Network Foundation Models (NFMs). Different from existing efforts, we focus on hidden representations analysis rather than pure downstream task performance and analyze NFMs through a three-part evaluation: Embedding Geometry Analysis to assess representation space utilization, Metric Alignment Assessment to measure correspondence with domain-expert features, and Causal Sensitivity Testing to evaluate robustness to protocol perturbations. Using five diverse network datasets spanning controlled and real-world environments, we evaluate four state-of-the-art NFMs, revealing that they all exhibit significant anisotropy, inconsistent feature sensitivity patterns, an inability to separate the high-level context, payload dependency, and other properties. Our work identifies numerous limitations across all models and demonstrates that addressing them can significantly improve model performance (up to 0.35 increase in $F_1$ scores without architectural changes).

## 1 Introduction

Machine learning solutions are widely applied in the networking domain, with numerous applications ranging from traffic classification [5, 32, 58, 59] and anomaly detection [28, 49] to quality of service optimization [6, 36, 57] and network security [2, 38]. However, many of them fail to generalize to production environments [8, 26, 34, 60] due to inherent biases in data collection methodology, limited coverage of operational scenarios, distribution shifts, and other factors, creating fundamental generalizability challenges.

**Network foundation models as a potential solution.** As a potential answer to these challenges, network foundation models (NFMs) have been gaining traction in the networking community [23, 32, 41, 52, 53, 59]. Similar to foundation models in other application domains, these models incorporate additional self-supervised pretraining phases that utilize unlabeled data to learn critical spatial, temporal, and causal relationships. NFMs represent a paradigm shift in network analysis, moving from the development of task-specific learning models to the use of general-purpose pretrained representations. Significant aspects of this shift include (i) generalizability: NFMs learn representations that transfer across multiple network analysis tasks; (ii) scale: NFMs process raw packet data and require no domain expert-based feature engineering; and (iii) impact: NFMs are expected to be increasingly deployed in production network systems for security, optimization, and monitoring. The adoption of NFMs promises to alleviate the difficulties caused by the growing complexity of modern network infrastructures and is viewed as an important step towards realizing the vision of self-driving networks.

**Diverse architectures complicate evaluation.** While aiming toward the similar goal of utilizing unlabeled data, design architectures and pretraining tasks vary significantly between existing NFMs,

---

[*]Corresponding author: rbeltiukov@ucsb.edu

39th Conference on Neural Information Processing Systems (NeurIPS 2025) Track on Datasets and Benchmarks.

including masked token prediction purely on raw network payload [32] or packet header bytes [37, 41, 52]; flow statistics calculation [23]; patched image reconstruction [53, 59]; and modifying foundation models developed for other domains and applying them to the networking domain [27, 51]. Resulting largely from the variability in network data preprocessing (see Appendix A for additional details), such diversity obscures the influence of the pretraining phase on model performance, making it difficult to understand what knowledge the model gains during pretraining. Historically, the community has relied on the performance of such models on fine-tuning tasks [40], which utilize small (compared to the pretraining phase) labeled datasets as quality indicators and comparison metrics between models. However, this approach poses challenges for understanding foundation models' design choices, pretraining tasks, chosen pretraining datasets, and overall knowledge gained during the critical pretraining phase.

**Beyond downstream tasks: intrinsic evaluation.** In this paper, we address these limitations and complement the existing downstream task-oriented research by exploring and assessing the representational quality of NFMs' embeddings without depending on fine-tuning problems. Specifically, we develop three complementary analysis techniques to answer specific questions: (1) Embedding Geometry Analysis (§3.1) quantifies *how effectively models distribute representations across the embedding space* through anisotropy analysis, measuring the entanglement of learned representations and their influence on model performance. (2) Metric Alignment Assessment (§3.2) evaluates *feature correlation to identify whether models capture domain-expert metrics* like flow duration or TCP window dynamics. (3) Causal Sensitivity Testing (§3.3) employs perturbation analysis to evaluate *how embeddings respond to controlled protocol and context modifications, revealing higher-order context understanding*, including traffic shaping policies and congestion control mechanisms.

**Key findings.** Our empirical results demonstrate (§4.3) that embeddings from publicly available pretrained models exhibit significant anisotropy (mean cosine similarity = $0.86 \pm 0.09$) and that addressing this issue leads to model performance improvement (up to 0.35 increase in $F_1$ scores). We notice (§4.4) significant alignment of models' embeddings with packet lengths, time-based features, packet flags, and even payload information (§4.5) despite its frequent encryption or absence in production environments, and show a direct correlation between anisotropy and high-level context in the models.

Our work shifts the focus from downstream task performance to intrinsic representational properties and sets the stage for further explorations into developing next-generation NFMs that can be expected to facilitate the creation of performant, generalizable, and robust ML-based solutions for disparate learning problems – a critical step toward realizing the ambitious goal of fully self-driving computer networks. Our fully reproducible code is available at `https://github.com/maybe-hello-world/demystifying-networks`.

## 2 Preliminaries

In this section, we provide information about existing evaluation efforts and preliminary information on what hidden context refers to in the networking domain.

### 2.1 Existing evaluation efforts

To our best knowledge, several papers attempted to taxonomize the existing works in the area of foundation models for networking. Bovenzi et al. [10] contains a survey of existing works in GenAI for networking, including both text-based and traffic-based approaches, and raises various questions about interpretability and efficiency, but does not provide any comparative analysis of the models and their performance. Wickramasinghe et al. [54] provides a comprehensive analysis of both feature-engineered and foundation model-based approaches for network traffic classification, including design choices, feature selection, traffic granularity, and benchmarks and downstream tasks used. The work also implements several data occlusion strategies to evaluate the influence of different features on model performance and includes model development guidelines, though these results are derived from a single dataset for two models only. Qian et al. [40] provides a downstream task-oriented framework for evaluating classic and pretrained models for network traffic classification, relying on 7 public datasets covering 20 existing tasks and 7 different models for evaluation (with only two foundation models). Despite including an extensive set of downstream results, this work does not investigate model design choices, input data, or embedding structure and quality.

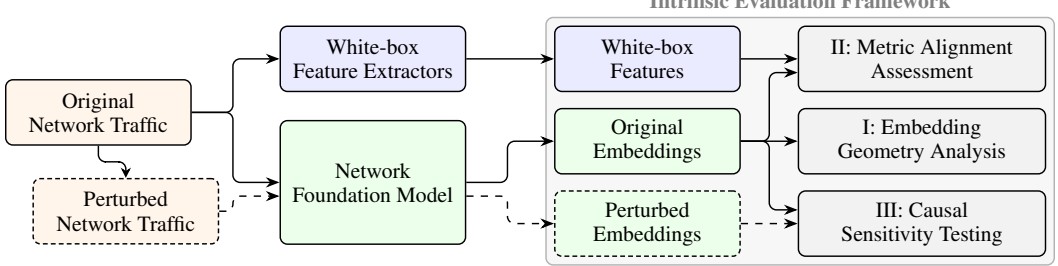

Figure 1: Visual overview of the proposed framework. Color indicates different stages of the analysis and dashed lines and boxes denote the perturbed network traffic path.

## 2.2 Hidden context in networking

One key learning challenge in the networking domain is that network traffic is influenced by various *hidden context $C$*, which is not explicitly expressed in the network traffic traces but significantly influences it. This context consists of: various **network conditions**, which include partially observable characteristics of the network (throughput, latency), traffic shaping policies, network congestion situation, and influence of other network traffic passing through the same interfaces; **application behavior**, which includes participants and their communication rules, such as communication protocols and congestion control algorithms, adaptive bitrate algorithms, application specific communication pattern (prevalence of download traffic over upload), inherent burstiness of the traffic; and others.

Both aspects of networking hidden context can often depend on one another, introducing *cross-dependencies* that have to be dealt with during analysis. Examples include how network conditions influence application behavior (e.g., ABR algorithm choices of bitrate [6, 7, 9, 36, 56]), and vice versa (e.g., TCP congestion control algorithms fairness [3, 35, 55]), or how even more complicated multi-way dependencies may arise (e.g., usage of performance-enhancing proxies [33] in geostationary satellites which terminate connections and introduce their own behavior on multiple levels).

## 2.3 Necessity of intrinsic evaluation

Since traditional evaluation efforts rely exclusively on downstream task performance, they suffer from critical shortcomings such as (i) performance scores provide no insight into what the models actually learn about network conditions or application behaviors, (ii) high performance for one task does not guarantee generalizability to new network domains, (iii) models can achieve good performance through dataset-specific shortcuts rather than genuine network understanding, and (iv) practitioners have no guidance for choosing models or identifying failure modes. Addressing these shortcomings calls for creating an intrinsic evaluation framework that can reveal what the models actually learn about network traffic, independent of specific tasks.

## 3 Intrinsic Evaluation Framework

NFMs pretrained on raw packet traces inherently encode an internal *latent space* reflecting various network and application characteristics captured from the utilized training data. To gain insights into these characteristics, we explore *embeddings* generated from diverse network data by *frozen* NFMs.

Using the calculated embeddings, we aim to: quantify how effectively models distribute representations across their available dimensions (§3.1), demonstrating if they capture both shared protocol patterns and distinctive flow characteristics; measure the correspondence between learned embeddings and established network metrics (§3.2), revealing whether NFMs inherently calculate the same statistical features that domain experts have engineered over decades; and evaluate how embeddings respond to controlled perturbations at both protocol and contextual levels (§3.3), verifying that models learn meaningful causal dependencies. For each technique, we provide a claim (*what NFMs should be able to do*), the rationale behind this claim, and our methodology to evaluate NFMs for the given claim. A visual overview of the Intrinsic Evaluation Framework is presented in Figure 1.

**Applicability.** In theory, our intrinsic evaluation framework can be used to evaluate any model that learns latent space representations, but it is most valuable for models that involve task-agnostic pre-training. While traditional approaches (e.g., neural networks trained using supervised learning techniques) can be evaluated using our metric alignment component, they often lack the explicitly extracted latent representations that make geometric analysis and causal sensitivity testing meaningful.

Importantly, because of the data representation challenges posed by network traffic (see Appendix A), how a model's input data is being preprocessed is intrinsically entangled with what model architecture is selected. Since both aspects reflect the explicit decisions that the original model's developers made, we are unable to evaluate them individually.

## 3.1 Embedding Geometry Analysis: quantifying representation space utilization

**Claim.** NFMs that effectively capture both shared protocol semantics and distinctive per-flow packet dynamics produce embeddings that efficiently utilize the full representation space. These models generate flow representations that distribute anisotropically throughout the latent space rather than collapsing into a concentrated cluster.

**Rationale.** Effective network modeling requires capturing both global protocol patterns and flow-specific variations. A well-distributed embedding geometry with measured anisotropy indicates [21] that the model distinguishes between flows along multiple meaningful dimensions. Such embeddings encode critical network characteristics including spatial (handshake vs. data exchange vs. teardown) or temporal patterns (short vs. long inter-arrival times) and causal reactions (packet loss vs. delays). Conversely, embeddings that cluster tightly with high cosine similarity suggest the model compresses diverse flows into nearly identical representations (similar to Li et al. [31], Su et al. [48]), overlooking subtle per-flow variations essential for robust network reasoning and downstream task performance.

**Methodology.** To quantify embedding geometry, we employ the established concept of anisotropy from contextualized language models [11, 16, 20], expressed by cosine similarity between different traffic flows. For each network traffic flow $x_j$, we extract its hidden representation $h_j$ from the final encoder layer of an NFM (as defined by the authors of the model). Across a set of $n$ such embeddings $\{h_j\}_{j=1}^n$, we estimate the anisotropy score $\mathcal{A} = \mathbb{E}_{i \neq j}[\cos(h_i, h_j)] \approx \frac{1}{|S|} \sum_{(i,j) \in S} \cos(h_i, h_j)$ (where $cos(u,v)$ is the $cos$ similarity), by sampling random flow pairs $S$ similar to Ethayarajh [17].

Further, we analyze the contributions to anisotropy of the largest dimensions by using Mean Cosine Contribution (MCC) from Hämmerl et al. [25] (defined for a dimension $k$ as $MCC(k) = \frac{1}{|S|} \sum_{(i,j) \in S} CC_k(h_i, h_j)$, where $CC_k(u,v) = \frac{u_k v_k}{||u|| \, ||v||}$) to ensure no single axis dominates. Lower values of $\mathcal{A}$ combined with a uniform MCC distribution indicate that embeddings uniformly utilize the available dimensions, possibly facilitating distinguishing between variations in network behavior.

## 3.2 Metric Alignment Assessment: measuring correspondence with domain-expert features

**Claim.** NFMs implicitly compute well-known network performance and behavioral metrics within their latent space. Flow-level embeddings encode critical network statistics such as flow duration, packet size distributions, and TCP dynamics without explicit supervision.

**Rationale.** Over decades, network domain experts have engineered various statistical white-box features and successfully used them for developing traffic-based machine learning solutuons [2, 30, 45]. By measuring the structural alignment between hidden representations of NFMs and these network metrics, we can verify whether models have learned traditionally meaningful generalizable network semantics rather than superficial correlations with particular downstream labels.

**Methodology.** We select a set of established network metrics calculated by CICFlowMeter [30], which have been extensively validated for traffic classification in prior works [2]. Let $\{x_j\}_{j=1}^n$ be our set of flows and let $m_i(x_j)$ denote the value of the $i$th CICFlowMeter metric for flow $x_j$. We extract the embedding $h_j \in \mathbb{R}^d$ from the NFM's final encoder layer before any task-specific heads (ET-BERT, netFound) or decoder layers (YaTC, NetMamba) and assemble $H = [h_1, \ldots, h_n]$.

For each metric $m_i$, we compute the similarity index $\rho(m_i, h_j)$ between the metric and the model's embedding representation across all traffic flows in the dataset using Centered Kernel Alignment (CKA) [29]. Unlike cosine similarity, CKA allows for calculating the similarity index between representations of different dimensionalities and is invariant to orthogonal transformations and

isotropic scaling, allowing for effective capture of both linear and non-linear relationships between representations. CKA values near 1 demonstrate that semantic probes successfully extract network-level metrics from embeddings, validating the model's implicit computation of these key features.

### 3.3 Causal Sensitivity Testing: interventional analysis of protocol and context dependencies

**Claim.** NFMs that effectively encode both protocol semantics and network context exhibit two key properties: (1) protocol-relevant perturbations produce quantifiable, focused changes in embedding similarity scores, and (2) high-level network contexts (e.g., congestion control algorithms, queue management policies) create distinct, linearly separable embedding subspaces. These properties are proof that the model learns meaningful causal dependencies between inputs and network conditions.

**Rationale.** A comprehensive test of causal understanding requires dual validation: we aim to verify both bottom-up causality (how low-level protocol features influence embeddings) and top-down contextual awareness (how high-level network conditions shape representation space). We explicitly separate these efforts into two complementary methodologies – feature perturbation analysis and context discrimination – to provide quantitative evidence of whether an NFM has learned a coherent causal model of network behavior spanning multiple levels of abstraction.

**Methodology: sensitivity to protocol-relevant perturbations.** For a selected dataset comprising flows $\{x_1, x_2, ..., x_n\}$, we compute the baseline similarity score $S(\{h_j\}_{j=1}^n)$, where $h_j$ denotes the embedding of the $j^{th}$ flow and $S(\cdot)$ quantifies the average pairwise cosine similarity between embeddings. We define a perturbation strategy $\delta(f)$ operating on a subset of features $f \subseteq F$, where $F = \{f_1, f_2, ..., f_n\}$ constitutes the complete set of traffic header features. For our analysis, we use a token replacement strategy that implements any token replacement from a valid range of tokens from the model's dictionary [43]. This strategy applies uniformly across all flows in the dataset.

Given a feature perturbation, we derive the perturbed flows $\{x_j^{\delta(f)}\}_{j=1}^n$ and extract their corresponding embeddings $\{h_j^{\delta(f)}\}_{j=1}^n$, and then measure the similarity score between perturbed embeddings $h_j^{\delta(f)}$ and the original $h_j$ using cosine similarity to measure changes in the hidden representations.

A significant decrease in the similarity score after perturbation indicates that the model's representation exhibits sensitivity to the perturbed features. Large changes for protocol-relevant features demonstrate that these features causally influence the model's internal representations. Such findings reveal that the model implicitly learns to encode protocol-relevant features, aligning with domain knowledge about network protocol behavior.

**Methodology: extraction of exogenous network context.** First, we use a network emulator to generate network traffic with distinct high-level contexts ($c_j \in \{1, \ldots, C\}$). Each context has multiple possible values, such as application type (video streaming vs. conferencing), congestion control algorithm (CUBIC vs. BBR), queue management policies (pFIFO vs. CoDEL), and cross traffic characteristics at the bottleneck link (more bursty and less intense or vice versa). For any given NFM, we extract embeddings ($h_j$) from the last encoder layer that represent the encoded state of $j^{th}$ flow. We compute these embeddings across different combinations of high-level contexts.

To quantify the NFM's context sensitivity, we employ two complementary analyses. First, we calculate pairwise cosine similarities between embeddings generated under different context combinations. We also establish a reference point by computing the average similarity score from embeddings of publicly available unlabeled datasets (the "average"). We then measure the relative change in similarity when moving from a base context (with default parameter values) to each alternative context combination. By comparing these changes against the deviation of the base context from the baseline average, we can contextualize the magnitude of embedding shifts. A significantly higher relative change in similarity compared to the baseline deviation suggests that the model effectively captures the distinctive characteristics of different network contexts.

As a second quantitative measure, we train a simple logistic regression classifier built on top of the frozen embeddings to distinguish between different contexts, and calculate its $F_1$ score. This approach assesses whether the information necessary to separate contexts is linearly accessible from the embedding space. A high $F_1$ score confirms that the NFM's embeddings internalize unobserved network conditions and can effectively discriminate between different operational scenarios, validating that the model learns meaningful representations of network behavior under varying conditions.

# 4  Benchmarking

In this section, we evaluate state-of-the-art NFMs using our proposed evaluation framework.

## 4.1  Network Foundation Models

Our benchmarking applies the intrinsic evaluation framework to diverse architectural approaches in NFMs. We selected four state-of-the-art representative NFMs that span different design philosophies, input representations, and pretraining strategies to evaluate how these fundamental choices impact representational quality.

For each model, we used publicly available pretrained checkpoints provided by the authors without modifications, maintaining each model's original data processing pipeline to ensure fair comparison. Our selection includes **YaTC** [59], **ET-BERT** [32], **netFound** [23], and **NetMamba** [53]. Additional information about their architectures can be found in Appendix B.

These models represent key design dimensions in NFM architecture: input modality (headers vs. payload vs. both), sequence length (5 vs. 60 packets), architectural foundation (Transformer vs. Mamba), and pretraining objectives (single vs. multi-task). Through our framework, we assess how these design choices influence embedding geometry, metric alignment, and causal sensitivity.

## 4.2  Datasets

Our benchmarking requires diverse network traffic datasets to ensure comprehensive evaluation across varying network conditions, traffic types, and operational settings. We selected five datasets spanning both controlled environments (**Android Crossmarket** [42], **CIC-IDS2017** [46], **CIC-APT-IIoT24** [22]) and real-world deployments (**CAIDA** [1], **MAWI** [13]). For preprocessing, we standardized all datasets by extracting uniform flow records using the data processing scripts provided by the authors of the considered models. We refer the reader to Appendix B for additional details.

## 4.3  Embedding Geometry Analysis

Table 1: Mean cosine similarity (cos) between embeddings and Mean Cosine Contribution (MCC) of the top dimension of the embeddings towards the average cosine similarity.

| Dataset | YaTC | | ET-BERT | | netFound | | NetMamba | |
|---|---|---|---|---|---|---|---|---|
| | $cos$ | $MCC$ | $cos$ | $MCC$ | $cos$ | $MCC$ | $cos$ | $MCC$ |
| **Crossmarket** | 0.85 | 0.25 | 0.88 | 0.02 | 0.69 | 0.01 | 0.93 | 0.02 |
| **CIC-APT-IIoT24** | 0.87 | 0.27 | 0.88 | 0.02 | 0.82 | 0.01 | 0.98 | 0.02 |
| **CIC-IDS2017** | 0.85 | 0.22 | 0.74 | 0.01 | 0.69 | 0.01 | 0.92 | 0.02 |
| **CAIDA** | 0.87 | 0.21 | 0.71 | 0.01 | 0.86 | 0.01 | 0.99 | 0.03 |
| **MAWI** | 0.88 | 0.19 | 0.78 | 0.01 | 0.94 | 0.02 | 0.99 | 0.02 |

Table 1 presents results from applying our embedding geometry analysis (additional results can be found in Appendix C).

**Models are not consistent in the anisotropy scores.** Our analysis reveals significant variability in how different NFMs utilize their representation space. netFound demonstrates more variability in space utilization (with anisotropy score varying between 0.69 and 0.94 for different datasets) compared to autoencoder-based models YaTC and NetMamba (which have more consistent scores of 0.85-0.88 and 0.92-0.99, respectively). ET-BERT occupies an intermediate position (0.71-0.88).

**MCC analysis reveals distinct failure modes.** While anisotropy scores identify models with concentrated representations, our Mean Cosine Contribution analysis further distinguishes between qualitatively different representational problems. NetMamba's uniform MCC values (the highest being around 0.03) indicate semantically collapsed representations without any single dimension contributing significantly to the collapse, while YaTC's uneven distribution (MCC 0.19-0.27) reveals problematic dimensional dominance where a single dimension captures disproportionate variance.

**Geometric properties predict environmental responses.** Embedding geometry analysis systematically exposes how different architectures respond to dataset variations. The observation that most

models produce higher cosine similarity (more collapsed representations) on real-world datsets — with ET-BERT uniquely showing the opposite pattern — provides predictive insight into how these models might generalize to deployment environments with different traffic distributions.

Table 2: $\Delta F_1$ of NetMamba after fine-tuning a single linear layer for 30 epochs on decorrelated embeddings (over five training runs).

|  | Crossmarket | CIC-IDS2017 | CIC-APT-IIoT24 |
|---|---|---|---|
| **NetMamba** | $+0.35 \pm 0.02$ | $+0.11 \pm 0.27$ | $+0.03 \pm 0.02$ |

**Anisotropy directly impacts performance.** While anisotropy is a measure that quantifies embedding quality, it does not directly capture how this geometry affects downstream performance [16]. Therefore, we complement our anisotropy analysis by considering an additional metric: isotropification gain. This metric quantifies the potential performance improvement when correcting for suboptimal embedding distributions. We apply a deterministic decorrelation transformation [24] to the frozen raw representations and retrain only a linear classifier for the downstream task. We define the isotropification gain as $\Delta F_1 = F_1^{\text{decorrelated}} - F_1^{\text{raw}}$. Additional popular techniques, such as batch normalization and whitening, are discussed in Appendix D.

To validate that embedding geometry directly affects model utility, we identified NetMamba as an ideal candidate for isotropification based on its high anisotropy. Table 2 shows the resulting $F_1$ score improvements (+0.03 to +0.35) after applying decorrelation transformations. Note that this improvement is calculated for a simplified classification head (linear classifier), and results for more complicated models can vary. However, these results suggest that anisotropy is not merely a descriptive metric but can predict potential performance gains that derive from applying representation enhancement techniques to NFMs.

## 4.4 Metric Alignment Assessment

Table 3: Averaged CKA similarity index between CICFlowMeter white-box features and model embeddings.

|  | Crossmarket | CIC-APT-IIoT24 | CIC-IDS2017 | CAIDA | MAWI | Average |
|---|---|---|---|---|---|---|
| **YaTC** | 0.098 | 0.148 | 0.092 | 0.014 | **0.070** | 0.093 |
| **ET-BERT** | 0.012 | 0.014 | 0.064 | 0.033 | 0.026 | 0.029 |
| **netFound** | **0.156** | **0.219** | **0.167** | **0.052** | **0.070** | **0.143** |
| **NetMamba** | 0.047 | 0.141 | 0.042 | 0.030 | 0.051 | 0.066 |
| **Average** | 0.078 | 0.131 | 0.091 | 0.032 | 0.055 | 0.077 |

Table 3 presents the averaged Centered Kernel Alignment (CKA) similarity index between model embeddings and CICFlowMeter features across all datasets, quantifying the degree to which NFMs implicitly encode established network metrics without explicit supervision. Appendix E provides a breakdown of this average similarity index into the similarity indices of the individual features.

**Architectural design determines metric alignment.** Our analysis reveals substantial variation in how different architectures encode domain-expert features. netFound consistently demonstrates the highest alignment (average CKA: 0.143), suggesting its multi-input approach and packet burst representation effectively capture statistical properties identified by domain experts. In contrast, ET-BERT's payload-only approach shows minimal alignment (average CKA: 0.029), indicating traditional network metrics cannot be reliably extracted from payload representations alone. Since YaTC and NetMamba share similar architectural designs, they demonstrate similar CKA performance across datasets.

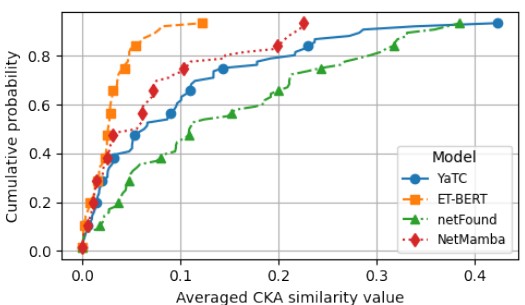

Figure 2: CDF of CKA similarity among different model embeddings and CICFlowMeter features averaged across all five datasets.

**Feature diversity enhances metric coverage.** Figure 2 shows the CDF of the CKA similarity index, revealing that models with diverse input features exhibit a more uniform alignment distribution compared to models with constrained input features (i.e., more slowly increasing CDF, resulting in more features having a high CKA similarity value). Models processing packet headers, payload, and flow metadata (netFound) capture a broader spectrum of expert-designed metrics compared to specialized architectures (ET-BERT, YaTC, NetMamba), indicating that architectural diversity directly impacts the comprehensiveness of implicit feature computation.

**Environmental context affects metric relevance.** All models demonstrate significantly lower CKA similarity on real-world datasets (CAIDA: 0.032, MAWI: 0.055) compared to datasets from controlled environments (CIC-APT-IIoT24: 0.131, CIC-IDS2017: 0.091). This consistent pattern suggests that existing NFMs struggle to encode established network metrics in production environments.

Table 4: Average CKA similarity index between model embeddings and CICFlowMeter features. Top 5 features with the highest similarity index are presented for each model.

| Model | Feature name | Mean CKA | CKA Std | Model | Feature name | Mean CKA | CKA Std |
|---|---|---|---|---|---|---|---|
| YaTC | Fwd Seg Size Min | 0.423 | 0.185 | netFound | FWD Init Win Bytes | 0.385 | 0.352 |
|  | FWD Init Win Bytes | 0.339 | 0.241 |  | Packet Length Max | 0.372 | 0.302 |
|  | Packet Length Min | 0.286 | 0.085 |  | Fwd Seg Size Min | 0.359 | 0.338 |
|  | SYN Flag Count | 0.282 | 0.230 |  | Fwd IAT Total | 0.331 | 0.173 |
|  | Bwd Packet Length Min | 0.272 | 0.209 |  | Flow Duration | 0.331 | 0.170 |
| ET-BERT | SYN Flag Count | 0.122 | 0.080 | NetMamba | Fwd Segment Size Avg | 0.225 | 0.244 |
|  | FWD Init Win Bytes | 0.081 | 0.068 |  | Fwd Packet Length Mean | 0.225 | 0.244 |
|  | Packet Length Std | 0.078 | 0.094 |  | Average Packet Size | 0.221 | 0.258 |
|  | FIN Flag Count | 0.074 | 0.074 |  | Packet Length Min | 0.215 | 0.273 |
|  | Packet Length Max | 0.068 | 0.097 |  | Packet Length Mean | 0.210 | 0.253 |

**Temporal-spatial features dominate alignment patterns.** Table 4 shows that all models demonstrate the highest alignment with features related to packet length, segment size, and flow burstiness. For instance, netFound shows particularly strong alignment with features *forward initial window bytes* (CKA: 0.385) and *packet length max* (CKA: 0.372), while YaTC prioritizes the feature *forward segment size min* (CKA: 0.423). Even ET-BERT, despite overall low alignment, shows relative preference for the feature *SYN flag count* (CKA: 0.122). These observations suggest that all models discover the importance of learning initial connection behavior and data transfer volumes.

### 4.5 Causal Sensitivity Testing: protocol-relevant perturbations

Table 5: Cosine similarity between the baseline and embeddings after perturbing IP and TCP features.

| | YaTC | | ET-BERT | | netFound | | NetMamba | |
|---|---|---|---|---|---|---|---|---|
| Baseline *cos* | 0.87 | | 0.71 | | 0.86 | | 0.99 | |
| | % tok | *cos* | % tok | *cos* | % tok | *cos* | % tok | *cos* |
| SEQ/ACK | 5% | 0.61 | 0% | - | 22% | 0.99 | 5% | 0.98 |
| IP Total Length | 0.6% | 0.88 | 0% | - | 5% | 0.99 | 0.6% | 0.99 |
| IP TTL | 0.6% | 0.88 | 0% | - | 5% | 0.98 | 0.6% | 0.99 |
| TCP Flags | 0.9% | 0.86 | 0% | - | 5% | 0.99 | 0.9% | 0.99 |
| TCP Window Size | 2.5% | 0.67 | 0% | - | 5% | 0.99 | 2.5% | 0.99 |
| Payload | 75% | 0.18 | 100% | 0.48 | 33% | 0.99 | 75% | 0.62 |

Table 5 presents results of our protocol-relevant perturbations testing on the CAIDA dataset, measuring how modifications to specific protocol features affect embedding stability. We selected the CAIDA dataset because it contains diverse, unfiltered, real-world Internet backbone traffic and allows therefore for a more realistic evaluation of model robustness than using controlled laboratory datasets.

**Perturbation methodology accounts for model differences.** Since the considered NFMs employ substantially different tokenization techniques, we applied perturbations to semantically meaningful packet features (e.g., TCP Window Size, IP TTL) rather than model-specific tokens. Consequently, the same packet feature modification affects different percentages of tokens across models as shown in the "% tok" columns of Table 5. This approach ensures fair comparison by focusing on model-independent protocol semantics rather than model-specific implementations.

**Models generally maintain stability under header perturbations.** For most model-feature pairs (netFound, NetMamba, some features of YaTC), the cosine similarity between embeddings before and after header perturbation exceeds the average internal similarity of the baseline dataset. These findings indicate that single feature modifications do not significantly influence model representations from an embedding geometry perspective, indirectly showing architectural stability for header processing.

**Selective feature sensitivity reveals architectural priorities.** YaTC demonstrates selective sensitivity to modifying transport layer features, particularly SEQ/ACK (0.61) and TCP Window Size (0.67), even though these modifications affect only small portions of the input (5% and 2.5% of tokens, respectively). This targeted sensitivity suggests architectural attention to specific protocol control signals, a pattern that is not evident in the other models, all of which maintain near-baseline similarity across all header modifications.

**Architectural differences in protocol processing.** ET-BERT shows no sensitivity to header modifications. This result is not surprising because the model's architecture deliberately ignores headers. netFound demonstrates remarkable stability (0.98-0.99) across all header modifications despite processing these fields, suggesting its representations prioritize higher-level patterns. These distinct responses directly reflect fundamental differences in how models process protocol information.

**Payload dependency across NFMs.** Unlike header features, payload perturbations introduce significant representation shifts across almost all models (YaTC: 0.18, ET-BERT: 0.48, NetMamba: 0.62). These results reveal a critical dependency on payload features for decision-making, even though this data is often encrypted or omitted in production environments. These findings highlight a potential model robustness concern that would go unnoticed in traditional downstream evaluations.

**Perturbation magnitude does not predict representation impact.** The percentage of tokens modified does not consistently correlate with representation changes. Small modifications to critical features (SEQ/ACK at 5% for YaTC) can produce larger representational shifts than more extensive modifications to other features (IP TTL at 0.6%), revealing an implicit hierarchy of feature importance within each model's internal representations.

### 4.6 Causal Sensitivity Testing: exogenous network context

We use the network emulator NetReplica [15] to generated five synthetic datasets, each consisting of $\approx 100$ network flows and obtained using specific choices of congestion control algorithms, AQM policies, and cross-traffic patterns. Additional details about NetReplica and the generated datasets can be found in Appendix J.

Table 6: Average *cos* similarity across all datasets, *cos* similarity change (w.r.t. difference between stability baseline and average value), and linear probing $F_1$ score for synthetic datasets.

| | YaTC | | ET-BERT | | netFound | | NetMamba | |
|---|---|---|---|---|---|---|---|---|
| Metric | *cos* | - | *cos* | - | *cos* | - | *cos* | - |
| Average | 0.8633 | - | 0.7977 | - | 0.8017 | - | 0.9639 | - |
| Stability baseline | 0.8939 | - | 0.9698 | - | 0.9700 | - | 0.9940 | - |
| | $\Delta cos$ | $F_1$ | $\Delta cos$ | $F_1$ | $\Delta cos$ | $F_1$ | $\Delta cos$ | $F_1$ |
| Congestion Control | -13.07% | 0.48 | 0.32% | 0.41 | -1.84% | **0.60** | 0.50% | 0.29 |
| AQM | 0.99% | 0.51 | -1.44% | 0.68 | -10.97% | **0.93** | -280.01% | 0.70 |
| Crosstraffic | -5.71% | 0.24 | 0.60% | 0.43 | -6.45% | **0.78** | 0.51% | 0.33 |
| All | 6.72% | 0.47 | -1.07% | 0.46 | -31.32% | **0.97** | -42.34% | 0.62 |

Table 6 presents average *cos* similarity values for a diverse set of public datasets used in this work ("Average" row) and for the stability baseline dataset (i.e., a fixed dataset with given parameters for all choices, selected as a baseline). Assuming that the average and stability baselines represent lower and higher bounds respectively for the possible similarity values for each model, we list for each different high-level context change (see rows of the table labeled "Congestion Control", "AQM", "Crosstraffic", and "All" for all changes) the $\Delta cos$ value which is calculated as the difference between the average *cos* similarity of the test dataset and the stability baseline. The reported $F_1$ scores denote the performance of a linear probe trained to separate between the stability baseline and the test dataset.

**All models are able to group the baseline traffic.** Table 6 ('Average' and 'Stability baseline' lines) shows that all models demonstrate much higher average $cos$ similarity (but staying below the upper bound provided by the stability baseline) compared to the average similarity in diverse datasets. This property suggest that the models are able to reliably group the similar traffic and is important, for example, for anomaly detection tasks, where previously unseen anomalous traffic should be separated from the normal traffic.

**High-level context insignificantly influences the $cos$ similarity of embeddings.** Given that average $cos$ similarity of diverse datasets and stability baseline are different for different models, we normalized $\Delta cos$ in Table 6 by the difference between the average $cos$ similarity of the stability baseline and average $cos$ similarity across diverse datasets, effectively measuring changes between the "lowest" and the "highest" measured $cos$ values. We notice that for all models, the $\Delta cos$ of the synthetic datasets is relatively small, implying that the resulting $cos$ is very close to the stability baseline. Except for NetMamba-AQM, NetMamba-All, and netFound-All, for all other model-dataset pairs, the absolute value of $\Delta cos$ is lower than 15%, an indication that the models do not explicitly disentangle the high-level context from the baseline. In some cases, the $\Delta cos$ is even positive (i.e., embeddings are grouped even closer), demonstrating that the model considers the traffic with the introduced changes to be more similar to the stability baseline than the stability baseline itself and does therefore not notice the high-level context change.

**$\Delta cos$ correlates with linear probing $F_1$ score.** For most of the model-dataset pairs, the lower average $cos$ score correlates with a higher linear probing $F_1$ score, demonstrating (similar to Table 2) that disentangling produced embeddings might lead to better fine-tuning performance even for more complicated fine-tuning tasks.

**Models are better at distinguishing different AQMs than different Congestion Control algorithms, crosstraffic patterns, or even all changes together.** For almost all models, the linear probing $F_1$ score is significantly higher for the AQM row in Table 6 compared to the Congestion Control and Crosstraffic rows. This observation suggests that the influence of AQM policies is easier captured from the raw network traffic, and sometimes even easier than when all changes are introduced together. Across all four rows, netFound has the highest $F_1$ scores, an indication that it is capable of capturing the high-level context changes. We also conducted additional experiments, including examining linear combinations of embeddings, and report the results in Appendix H.

# 5    Conclusion

In this paper, we introduce an initial approach to a principled evaluation of NFMs through intrinsic representation analysis and conduct a series of experiments on four SOTA NFMs over five diverse datasets for embedding geometry analysis, metric alignment assessment, and causal sensitivity testing. Our work reveals a number of critical insights, such as (1) all analyzed NFMs exhibit significant anisotropy that directly impacts downstream performance; (2) each model's architectural design fundamentally determines which network metrics can be reliably extracted from its representations; and (3) all models demonstrate concerning sensitivity to payload information despite its frequent encryption (or omission) in production environments. In particular, our proposed framework provides the network community with systematic tools for comparing different NFM architectures objectively, identifying model limitations before deployment, guiding architectural improvements based on intrinsic properties, and understanding failure modes that downstream metrics miss. At the same time, our findings highlight fundamental limitations in current NFM architectures and motivate future research on relationships between architectural choices, representation properties, and models' performance.

**Limitations.** The current work poses a number of limitations that affect the impact of our study. Our current scope of exploration and metrics used are far from exhaustive and are an initial attempt at evaluating NFMs' performance and latent knowledge without focus on downstream performance. Our application of the cosine similarity metric can produce incorrect insights [47] if applied to other models (e.g., which used cosine-based loss functions during pretraining). Finally, our dataset selection is based on publicly available data that can be flawed [34] or biased in terms of the network traffic patterns contained in the data.

## Acknowledgments and Disclosure of Funding

This work was supported in part by the National Science Foundation (CAREER Award No. 2443777 and CNS Award No. 2323229) and a research gift from Cisco. This research used resources of the National Energy Research Scientific Computing Center, a DOE Office of Science User Facility supported by the Office of Science of the U.S. Department of Energy under Contract No. DE-AC02-05CH11231 using NERSC award NERSC DDR-ERCAP0029768.

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

# A   Data representation for machine learning in networks

Network traffic (data exchanged between devices) is one of the main data sources for machine learning in networking. One of the unique characteristics of network traffic is that the same data is often represented in multiple ways simultaneously, such as raw packet captures that contain transmitted headers and payload (PCAPs), flow records and statistics summarizing each individual flow (finished communication session) between two endpoints (e.g., Zeek entries [39], IPFIX [4], NetFlow [14], CICFlowMeter [30], NTLFlowLyzer [45], etc.), and high-level extracted features (e.g., TCPInfo).

Being the most common representation, raw PCAPs contain all packets exchanged between the endpoints, including headers and payload. However, due to the nature of network communication and privacy concerns, the payload is often encrypted (e.g., TLS) and not available for analysis. Moreover, the volume of network traffic is often so large that it is difficult to analyze everything. In such cases, a practical solution is to analyze only a small portion of each network flow (e.g., the first few packets). These and other challenges also limit the availability of labeled datasets, making the use of self-supervised pretraining techniques especially attractive.

# B   Network Foundation Models and Datasets

**ET-BERT** [32] is a BERT-based network foundation model. ET-BERT uses only (encrypted) payload data of the first five packets of network traffic flows, ignoring packet headers, and is trained using masked token prediction and packet order prediction tasks. ET-BERT produces 768-dimensional embeddings.

**netFound** [23] is a transformer-based network foundation model that uses packet headers, payload, and additional flow information (such as packet interarrival times or total bytes transferred), extracting *packet burst* representations (up to 60 packets), potentially from different parts of the traffic flow. In addition to masked token prediction, netFound uses multiple additional tasks, such as metadata prediction or packet order prediction, to pretrain the model. The used checkpoint of netFound (large) produces 1024-dimensional embeddings.

**YaTC** [59] and **NetMamba** [53] are masked autoencoder-based network foundation models (based on Transformer and Mamba architectures), which both transform existing network traffic data into a fixed-size image-like representation. Both models use the first five packets of each network traffic flow, taking into account both headers and payload data. The two models produce 192 and 256-dimensional embeddings, respectively.

**Endogenous Datasets.** These datasets were generated in controlled settings with full ground-truth information:

- **Android Crossmarket** [42] contains around 66k flows from 215 Android applications across US, Chinese, and Indian app stores, capturing diverse mobile application traffic patterns during actual user interactions.

- **CIC-IDS2017** [46] is a benchmark intrusion detection dataset with 2.8M flows including both benign traffic and common attack patterns (e.g., brute force, DDoS, port scans, Heartbleed).

- **CIC-APT-IIoT24** [22] contains 3.7M flows representing 25 distinct advanced persistent threat techniques alongside benign traffic and is valuable for testing model sensitivity to subtle attack patterns.

**Real-World Traffic Datasets.** These datasets were collected from actual production networks without artificial manipulation:

- **CAIDA** [1] contains 1.5 million anonymized packet-level flows from an Internet backbone link between Los Angeles and San Jose (collected in 2023).

- **MAWI** [13] comprises 1 million randomly sampled flows from a major Internet Exchange Point in Tokyo (collected in 2023), capturing international transit traffic with significant geographic routing characteristics.

## C   Anisotropy of the embedding space

Table 7: Anisotropy of the embeddings with top dimensions (TD) and their Mean Cosine Contribution (MCC).

| Dataset | YaTC Anisotropy | TD | MCC | ET-BERT Anisotropy | TD | MCC | netFound Anisotropy | TD | MCC | netMamba Anisotropy | TD | MCC |
|---|---|---|---|---|---|---|---|---|---|---|---|---|
| | | 129 | 0.246 | | 732 | 0.017 | | 1008 | 0.009 | | 94 | 0.023 |
| Crossmarket | 0.85 | 137 | 0.051 | 0.88 | 98 | 0.012 | 0.69 | 282 | 0.008 | 0.93 | 175 | 0.021 |
| | | 146 | 0.034 | | 434 | 0.011 | | 816 | 0.007 | | 165 | 0.021 |
| | | 129 | 0.212 | | 732 | 0.010 | | 537 | 0.014 | | 94 | 0.026 |
| CAIDA | 0.87 | 137 | 0.057 | 0.71 | 527 | 0.010 | 0.86 | 282 | 0.014 | 0.99 | 165 | 0.024 |
| | | 146 | 0.043 | | 434 | 0.010 | | 816 | 0.009 | | 175 | 0.022 |
| | | 129 | 0.270 | | 732 | 0.018 | | 309 | 0.010 | | 94 | 0.025 |
| CIC-APT-IIoT24 | 0.87 | 137 | 0.039 | 0.88 | 98 | 0.012 | 0.82 | 1008 | 0.008 | 0.98 | 37 | 0.022 |
| | | 99 | 0.033 | | 434 | 0.011 | | 537 | 0.008 | | 222 | 0.021 |
| | | 129 | 0.215 | | 527 | 0.012 | | 537 | 0.010 | | 94 | 0.025 |
| CIC-IDS2017 | 0.85 | 137 | 0.055 | 0.74 | 732 | 0.012 | 0.69 | 282 | 0.009 | 0.92 | 165 | 0.022 |
| | | 146 | 0.038 | | 434 | 0.011 | | 816 | 0.008 | | 175 | 0.020 |
| | | 129 | 0.191 | | 527 | 0.013 | | 282 | 0.016 | | 165 | 0.025 |
| MAWI | 0.88 | 137 | 0.054 | 0.78 | 434 | 0.011 | 0.94 | 537 | 0.016 | 0.99 | 94 | 0.025 |
| | | 146 | 0.044 | | 130 | 0.010 | | 354 | 0.010 | | 175 | 0.023 |

Table 7 provides anisotropy for each model and dataset separately, including information for the top three dimensions and their Mean Cosine Contributions towards anisotropy.

Table 8: MCC distribution statistics across all datasets.

| Model | Range | Std | Skew | Kurtosis | Gini |
|---|---|---|---|---|---|
| YaTC | 0.2105 | 0.0163 | 10.7569 | 129.9826 | 0.7522 |
| ET-BERT | 0.0157 | 0.0017 | 3.3156 | 16.2447 | 0.6547 |
| netFound | 0.0119 | 0.0012 | 3.5168 | 19.9052 | 0.6504 |
| NetMamba | 0.0250 | 0.0049 | 2.1100 | 4.4845 | 0.6388 |

Table 8 lists MCC distribution-related statistics across all datasets, providing additional insight into the variability of the MCC metric.

## D   Comparison of BatchNorm layer, decorrelation, and whitening

Table 9: $\Delta F_1$ test score after finetuning for 30 epochs on frozen embeddings with BatchNorm layer (BN), decorrelation (D), or whitening (W).

| | Crossmarket | | | CIC-IDS2017 | | | CIC-APT-IIoT24 | | |
|---|---|---|---|---|---|---|---|---|---|
| | BN | D | W | BN | D | W | BN | D | W |
| **YaTC** | +0.013 | +0.025 | +0.006 | -0.009 | +0.002 | -0.003 | +0.007 | +0.001 | -0.018 |
| **ET-BERT** | +0.007 | +0.002 | +0.008 | +0.010 | +0.014 | +0.017 | +0.008 | +0.038 | +0.021 |
| **netFound** | +0.016 | +0.064 | +0.073 | +0.001 | +0.001 | +0.001 | +0.047 | +0.053 | +0.064 |
| **NetMamba** | +0.129 | +0.350 | +0.228 | -0.052 | +0.112 | +0.031 | +0.029 | +0.033 | +0.008 |

To further explore the influence of various normalization techniques, we implemented three different techniques (batch normalization, decorrelation, and whitening) and applied them to all models for all datasets containing downstream task labels. We trained a single linear layer for 30 epochs on the produced embeddings. The results are presented in Table 9 and show that the decorrelation technique consistently provides the largest increase in the $F_1$ test scores among all implemented techniques. At the same time, we also notice that the results can be expected to depend on the considered model-dataset combination.

## E   CICFlowMeter feature correlation

Figure 3 provides an overview of the CKA similarity index for each feature produced by CICFlowMeter. The CKA values are averaged over all datasets. We notice that ET-BERT shows a lack of correlation between (payload-only) embeddings and the flow features, while YaTC and netFound clearly correlate with various features, some of which might be explicitly connected to their training algorithm and loss functions.

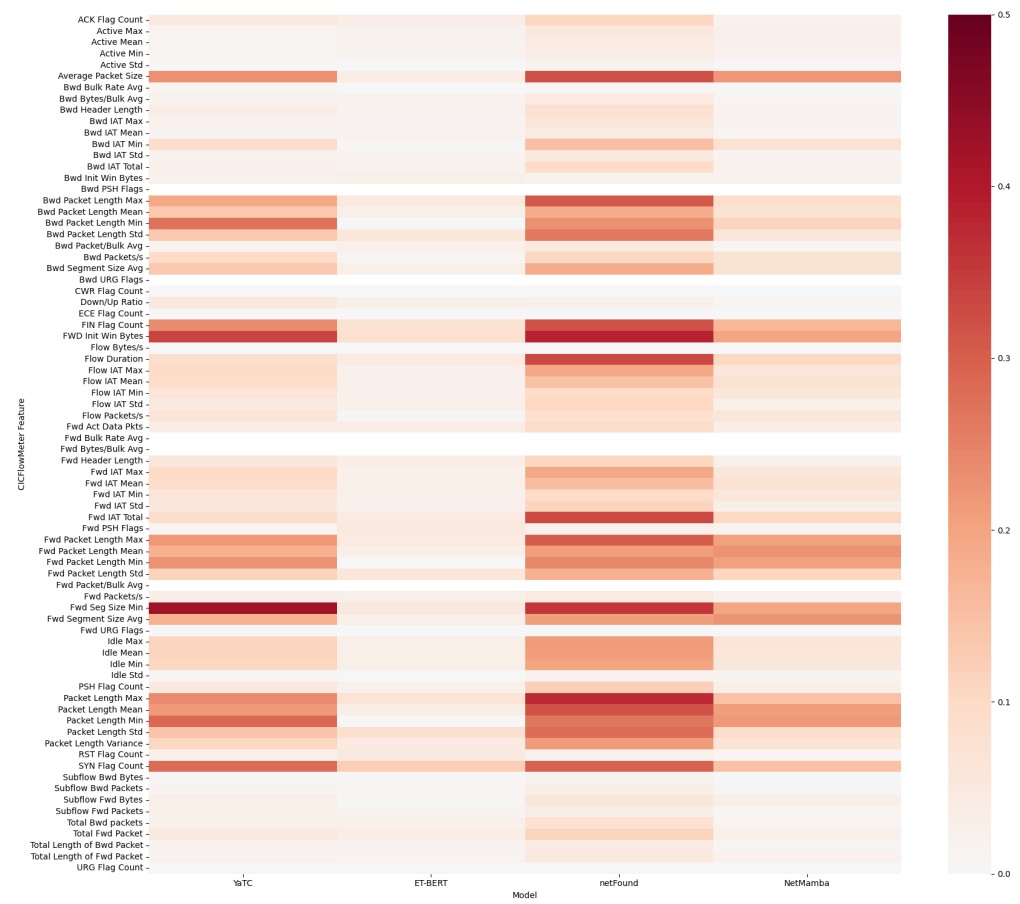

Figure 3: Similarity index of each of CICFlowMeter features per model. YaTC and netFound demonstrate higher similarity with well-known white-box features compared to other models.

# F  Manifold dimensionality

Intrinsic Dimension (ID) calculation is a common technique [12, 50] for evaluation of the effective degrees of freedom in the embedding space and efficiency of its usage. By comparing the intrinsic dimensions of the embedding space produced by different models on the same datasets, we can evaluate to what extent the models are able to capture the hidden context presented in the data. We used the TwoNN estimator [19] to calculate the ID of the embeddings produced by our models on all datasets. The results are presented in Table 10.

Table 10: Manifold ID of the embeddings produced by the network foundational models.

|                  | YaTC  | ET-BERT | netFound | NetMamba |
|------------------|-------|---------|----------|----------|
| **Crossmarket**  | 7.36  | 108.62  | 6.07     | 7.85     |
| **CAIDA**        | 5.92  | 108.67  | 5.19     | 6.50     |
| **CIC-APT-IIoT24** | 11.58 | 131.66  | 2.83     | 0.60     |
| **CIC-IDS-2017** | 6.20  | 108.31  | 8.09     | 7.44     |
| **MAWI**         | 4.81  | 106.94  | 7.80     | 7.63     |

**Models' manifold ID is stable between datasets and similar between models.** Most of the models have similar manifold IDs between different datasets (except CIC-APT-IIoT24), indicating that the datasets contain approximately similar amounts of information extractable by the models, irrespective of whether the dataset is synthetic (collected in a controlled environment) or realistic (collected in a

real-world production network). Also, all models (except ET-BERT) are able to extract approximately the same amount of information from these datasets.

**ET-BERT manifold ID represents an architectural choice.** ET-BERT demonstrates much higher manifold ID values compared to all other models. This observation can be explained by the model's architectural choice of selecting only the payload for the embeddings computation. Since the payload is often encrypted and contains therefore in theory a uniformly distributed set of all possible input tokens, the model needs a much higher number of intrinsic dimensions to fully express the data.

## G  Zero shot clusterization

Silhouette score [44] is a common technique for evaluating the clustering quality of the embeddings using downstream task labels, with values ranging from $-1$ (meaning possibly wrong clusterization) to $+1$, where higher is better.

We calculated the Silhouette score on the Crossmarket and CIC-IDS-2017 datasets for all models and present the results in Table 11.

Table 11: Silhouette (clustering quality) score of network foundational models.

|  | YaTC | ET-BERT | netFound | NetMamba |
|---|---|---|---|---|
| **Crossmarket** | -0.1387 | -0.0551 | -0.3608 | -0.5043 |
| **CIC-IDS-2017** | -0.0226 | 0.0972 | -0.0127 | -0.0121 |

**Most of the embeddings have a negative Silhouette score.** We observe that for both datasets and all models (except ET-BERT and CIC-IDS-2017), the resulting score is negative. This finding suggests that the models' extracted embeddings are grouped according to different patterns and similarities compared to those provided by the datasets' authors. It also shows that embeddings extracted from the frozen pretrained models might not provide enough information for the downstream tasks and that additional unfreezing and training of the models might be required to improve their downstream-related extraction patterns.

## H  Embedding space linear transformation.

Inspired by Ethayarajh et al. [18], we designed an experiment to evaluate the linear transformation of the embedding space in networking foundation models. We used the same networking datasets as in subsection 4.6, where we use network traffic with a certain Congestion Control algorithm, AQM policy, and crosstraffic patterns for the baseline $E_{Base}$, and similar kind of traffic but with a different Congestion Control algorithm, AQM policy, and crosstraffic pattern for $E_{CC}$, $E_{AQM}$, and $E_{Cross}$ respectively. We also define $E_{All}$ for the network traffic with all the different Congestion Control algorithms, AQM policies, and crosstraffic patterns from the baseline.

To investigate the linear transformation properties, we define the vector $E'_R = E_{CC} + E_{AQM} + E_{Cross}$. As each of $E_{CC}$, $E_{AQM}$, and $E_{Cross}$ contains a single high-level context change and two unchanged contexts, to bring the resulting vector $E'_R$ to $E_{All}$, we subtract $2 * E_{Base}$ from $E'_R$. The

Table 12: $cos$ similarity and $L_1$ distance from the resulting embedding $E_R$ to embeddings of network data with various high-level context.

|  | YaTC | | ET-BERT | | netFound | | NetMamba | |
|---|---|---|---|---|---|---|---|---|
|  | $cos$ | $L_1$ | $cos$ | $L_1$ | $cos$ | $L_1$ | $cos$ | $L_1$ |
| $E_R$ to $E_{Base}$ | 0.9360 | 21.1 | 0.9833 | 114.6 | 0.9448 | 169.0 | 0.9955 | 25.3 |
| $E_R$ to $E_{CC}$ | 0.9385 | 20.9 | 0.9846 | 109.4 | 0.9574 | 137.1 | 0.9958 | 65.1 |
| $E_R$ to $E_{AQM}$ | **0.9445** | **20.3** | 0.9821 | 118.8 | **0.9617** | 129.7 | 0.9134 | 44.2 |
| $E_R$ to $E_{Cross}$ | 0.9385 | 21.1 | **0.9853** | **108.5** | 0.9556 | 145.2 | **0.9960** | **24.7** |
| $E_R$ to $E_{All}$ | 0.9408 | 20.4 | 0.9823 | 117.7 | 0.9596 | **125.9** | 0.9830 | 28.1 |

resulting vector $E_R$ is then defined as $E_R = E_{CC} + E_{AQM} E_{Cross} - 2 * E_{Base}$ and should be close to $E_{All}$ if the linear transformation property holds.

Table 12 shows the $cos$ similarity and $L_1$ distance from the centroid of the resulting embedding $E_R$ to the centroids of the embeddings $E_{Base}$, $E_{CC}$, $E_{AQM}$, $E_{Cross}$, and $E_{All}$. Higher $cos$ similarity and lower $L_1$ distance indicate the closest embeddings to the resulting embedding $E_R$.

**Neither of models seem to uphold the linear transformation property in both metrics.** Among the four models, neither of the models managed to demonstrate the desired result of $cos$ similarity being the highest between $E_R$ (resulting embedding after linear transformation) and $E_{All}$ (target embedding), instead often aligning $E_R$ more closely with $E_{AQM}$ or $E_{Cross}$. At the same time, $E_R$ produced by netFound has the lowest $L_1$ distance to $E_{All}$, showing that the model is able to express (to some extent) the required linear transformation properties and contains correct internal mapping of network context in the embedding space.

In addition, both the YaTC and netFound models have the lowest $cos$ similarity and the highest $L_1$ distance between $E_R$ and $E_{Base}$ (original embedding before any changes or transformations). This result highlights that both YaTC and netFound can notice high-level network context changes introduced by Congestion Control, AQM, and crosstraffic variations, distancing the resulting embedding from the original traffic even if this distance does not uphold linear transformation properties (in case of YaTC).

# I  Compute resources.

All experiments for this paper were conducted on a single node with 4 GPUs A100 with 80 GB GPU RAM. The server contained 512 GB of RAM and attached network storage for data storage purposes. Raw precalculated embeddings from all the models require around 40 GB of storage, and the full datasets require 200 GB of free space for storage.

A subset of the experiments (all experiments using precalculated embeddings) can be executed on a single CPU-based node (without GPU accelerators), but the execution time might vary.

# J  NetReplica

NetReplica [15] is a programmable network emulator that offers *realism* (i.e., captures complex protocol dynamics and application behavior in the generated data), and *controllability* (i.e., provides tunable knobs to control the network conditions for data generation). The emulator was originally designed to address the prevalent domain adaptation problem in networking, where models trained in (skewed) controlled settings fail to generalize in challenging production settings.

**System architecture.** NetReplica decomposes a complex network environment into one or more bottleneck links (i.e., links that experience congestion) and facilitates explicitly specifying each of the links' critical attributes to systematically create different types of network conditions. The system is built around three core abstractions: `Link(type, node1, node2)` for defining network paths, `Bottleneck(type, link)` for configuring constrained paths, and `CrossTraffic` for managing background traffic patterns. The system supports multiple backends, including LibreQoS, tc, and mahimahi for implementing traffic control policies.

Specifically, NetReplica exposes two categories of control knobs that allow researchers to precisely specify network conditions for a bottleneck link: (1) **Static attributes** provide levers for specifying fundamental network conditions, including link capacity (e.g., 10 Mbps vs. 100 Mbps), maximum queue length (affecting bufferbloat), traffic shaping policy (e.g., token bucket or leaky bucket), and active queue management policy (e.g., Random Early Detection, Controlled Delay). (2) **Dynamic attributes** account for the behavior of the cross-traffic that traverses bottleneck links and leverages endogenously generated packet traces collected in situ from real production networks. These traces are preprocessed into different cross-traffic profiles (CTPs), each characterized by metrics measuring traffic intensity (average throughput), burstiness (temporal variation), and heterogeneity (traffic composition).

**Data-generation setup.** We use three servers (A, B, C) with 2.2 GHz Intel Xeon processors, 192 GB RAM, and dual Intel-X710 10 Gbps NICs – one connected to the Internet via our campus network,

the other to a Cisco SX550X 10 Gbps switch. We configure the static and dynamic attributes for the bottleneck link that connects Server A and C (via Server B). We deploy video streaming (Puffer) and speedtest (NDT) servers on Server C with clients inside Docker containers on Server A. We use `tcpreplay` for replicating cross-traffic that offers precise timing control to preserve burst characteristics and temporal patterns during replay. The LibreQoS implementation at Server B uses an XDP-based bridge that provides efficient packet processing with minimal overhead.

**NetReplica-generated dataset.** We use a combination of control applications at endpoints (i.e., Server A and C) and static and dynamic attributes for the bottleneck link (LibreQoS at Server B) to synthesize different networking contexts. Each contextual subset contains $\approx 100$ network flows collected from the environment with fixed choices and small introduced variability in insignificant network metrics (e.g., minimal packet arrival time variation). In particular, this dataset contains the following subsets:

- **Stability baseline.** A subset of flows with BBR (as a congestion control algorithm), FIFO (as Active Queue Management), and cross-traffic profile #36.
- **Congestion Control.** A subset of flows with Cubic, FIFO, and cross-traffic profile #36.
- **AQM.** A subset of flows with BBR, CoDEL, and cross-traffic profile #36.
- **Crosstraffic.** A subset of flows with BBR, FIFO, and cross-traffic profile #29.
- **All.** A subset of flows with Cubic, CoDEL, and cross-traffic profile #29 (containing all three changes compared to the baseline).

The cross-traffic profiles #29 and #36 were purposefully selected to represent different combinations of intensity and burstiness. Profile #36 exhibits moderate intensity with low burstiness, while profile #29 features higher intensity with moderate burstiness, creating distinctly different network conditions. Each profile maintains consistent host counts and traffic composition to isolate the effects of temporal traffic patterns.

