# OpenReview forum: "Demystifying Network Foundation Models"
_NeurIPS.cc/2025/Datasets_and_Benchmarks_Track — NeurIPS 2025 Datasets and Benchmarks Track poster_

### Official Review · Reviewer_fUnR · 2025-06-21

**Rating:** 4
**Confidence:** 3

**Summary:**

This paper investigates the properties of network foundation models from the perspective of latent representations. It conducts a comprehensive analysis through three types of evaluations: embedding geometry analysis to assess the utilization of the representation space, metric alignment assessment to measure the correspondence with domain-specific expert features, and causal sensitivity testing to evaluate robustness against protocol perturbations. The authors conduct extensive experiments and performance analyses on existing state-of-the-art network foundation models.

**Dataset Code Accessibility:**

Yes

**Dataset Code Comments:**

The author provided the complete code.

**Ethical Considerations:**

No, there are no or only very minor ethics concerns

**Final Justification:**

Thanks for the authors' reply. My concerns have been solved, so I raised my score.

**Limitations Weaknesses:**

1. As to Embedding Geometry Analysis, the use of MCC is very strange. It can only measure the similarity of the largest dimension, which may not fully reflect the geometry of the embedding space. Perhaps the author could consider measuring the maximum and minimum differences in similarity among multiple dimensions? And the author lacks more explanations for the MCC values, such as a more detailed analysis of NetMamba. I can't understand why MCC=0.02 is regarded as collapsed.
2. As to Metric Alignment Assessment, the explanation could be more precise. For example, in an encoder-decoder architecture, is the metric applied to the final layer of the encoder? How is this handled in models that only use a decoder? Clarifying these points would strengthen the evaluation’s credibility.
3. Following Weakness 2, Figure 1 is also somewhat confusing. The paper should provide a more detailed explanation of the coordinate axes and clearly indicate how and where feature diversity is manifested.
4. As the author mentioned, they only analyzed some internal features, but the relationship between the features and downstream tasks is more important. A better evaluation criterion needs to prove that the performance of downstream tasks can be partially reflected through features.

**Strengths Contributions:**

1 I believe analyzing models from the perspective of latent representations is both meaningful and challenging. It offers deeper insights into the underlying factors driving model performance, potentially at a theoretical level.

2. The paper provides a comprehensive analysis by evaluating latent representations from three distinct perspectives.

3. The writing is clear and well-structured. By adopting a claim–rationale–methodology format, the paper effectively guides readers to understand the significance and approach of each proposed evaluation perspective.

---

> ### Author Rebuttal · Authors · 2025-07-31
>
> Thank you for your detailed technical review.
>
> We appreciate your recognition that analyzing models from the perspective of latent representations is both meaningful and challenging, and that our paper provides "a comprehensive analysis with clear and well-structured writing.”
>
> Below, we consider each of your technical concerns and provide a detailed account of how we will address them and improve the overall presentation.
>
> 1. MCC interpretation and dimensional analysis.
>
> We acknowledge the limitations of our use of MCC as part of the Embedding Geometry Analysis and the need for more clarity and precision with respect to the notion of “embedding collapse.”
>
> Our MCC calculation measures the mean contribution of each dimension rather than a single one, but for brevity, Table 1 contains only the top (most contributing) dimension. Table 7 in Appendix C provides an extended MCC analysis, including both MCC and top dimensions (TD) metrics across all models and datasets, and the source code contains calculations for all dimensions.
>
> We will add a more detailed analysis of NetMamba in the paper, addressing the combination of very high anisotropy and low skewness of the mean contribution of each dimension. We will also change the wording about "collapse" of the representation to avoid confusion between "embedding collapse" (when embeddings occupy a low-dimensional subspace) and "semantic collapse" (when MCCs of each dimension are not skewed but embeddings themselves are highly concentrated within a very small region).
>
> To further validate these results, your suggestion to examine "maximum and minimum differences in similarity among multiple dimensions" motivated us to perform additional experiments/analyses to provide more information about the dimensional utilization patterns. In particular, we computed different statistics (standard deviation, skewness, kurtosis, and Gini coefficient) related to the variability in inter-dimensional similarity. The results we obtained are consistent with the current MCC findings (YaTC exhibiting the highest inequality between dimension contributions, and NetMamba demonstrating the most uniform distribution), and we will include them in the paper.
>
>
> 2. Metric alignment assessment methodology
>
> We will clarify that our methodology for metric alignment assessment extracts features from the final representation layer before any task-specific heads. In particular, for transformer models (ET-BERT, netFound), we extract embeddings from the final layer of encoders (before any heads); for masked autoencoder models (YaTC, NetMamba), we consider the final encoder layer embeddings before the decoder. In addition, Figure 2 in Appendix E provides a detailed per-feature correlation analysis between our extracted embeddings and CICFlowMeter features.
>
> 3. Figure 1 enhancement
>
> Figure 1 shows the Cumulative Distribution Function (CDF) of CKA similarity values between model embeddings and CICFlowMeter features (approximately 80 features), but we acknowledge that it's unclear which dataset this analysis corresponds to. Based on your comment, we will enhance this figure by (i) clearly specifying which dataset(s) the similarity distribution represents, (ii) providing axis labels that clearly indicate CKA similarity values (x-axis) and cumulative probabilities (y-axis), (iii) providing dataset information in the caption to clarify the scope of analysis, and (iv) including annotations that show how different curve shapes indicate varying degrees of alignment with domain expert features.
>
> 4. Intrinsic features and downstream performance connection
>
> Your concern about demonstrating the relationship between intrinsic features and downstream performance highlights an important evaluation methodology question that requires clarification.
>
> In particular, our framework intentionally decouples intrinsic representation quality from downstream task performance. The reasons for this decoupling include (i) there exists an extensive literature on exploring and evaluating downstream performance (including networking area), (ii) traditional downstream performance evaluation conflates multiple factors (e.g., input modalities, pretraining architecture, pre-training data quality, fine-tuning datasets, model heads, fine-tuning hyperparameters, etc), making it challenging to isolate what NFMs actually learn about network conditions, protocols and traffic patterns; (iii) a model might achieve high downstream performance through dataset-specific shortcuts, aggressive fine-tuning, or advantageous input preprocessing, while still having limited intrinsic understanding of network semantics; and (iv) our main goal is to understand what NFMs learn about networking fundamentals, rather than optimizing performance on specific benchmarks, and provide insights that are intrinsic to and remain valid across different applications, input modalities, and fine-tuning procedures.
>
> Please note that the results for the downstream experiments reported in Table 8 in Appendix D serve a very specific and limited purpose: demonstrating the causal effect of improving geometric representation quality on task performance. These results show that addressing geometric deficiencies through decorrelation leads to measurable F1 score improvements, validating that intrinsic geometric properties have practical consequences. However, we deliberately avoid using these downstream improvements as the primary validation of our framework, since that would conflate intrinsic representational quality with the various above-mentioned factors that influence task performance.

---

### Official Review · Reviewer_jicE · 2025-06-25

**Rating:** 4
**Confidence:** 3

**Summary:**

This paper introduces three intrinsic evaluation frameworks, including embedding geometry analysis, metric alignment assessment, and causal sensitivity testing. Next, the authors evaluate the state-of-the-art NFMs using the proposed framework. The findings offer concrete insights, including the pervasive anisotropy in NFM embeddings, the varying metric alignment across different architectures, and a critical dependency on payload features.

**Dataset Code Accessibility:**

Yes

**Dataset Code Comments:**

The dataset code is readily accessible, well-documented and available in its final form in a usable format.

**Ethical Considerations:**

No, there are no or only very minor ethics concerns

**Final Justification:**

The authors' responses have addressed my concerns. I will keep the score.

**Limitations Weaknesses:**

For other network models rather than NFM, I'm wondering whether the proposed framework can be applied. Besides, I'm wondering whether the proposed framework is necessary for evaluation. I think a more detailed introduction on NFM and its significance/application is necessary, as well as the comparison with other network models.

**Strengths Contributions:**

This paper proposes a new method for evaluating Network Foundation Models (NFMs), which focuses on analyzing the hidden representations within the model rather than solely focusing on the performance of downstream tasks. The study constructed a three part evaluation framework: embedded geometric analysis, metric alignment evaluation, and causal sensitivity testing. Through the evaluation of four mainstream NFM models on different datasets, the paper found that these models generally have problems such as anisotropy, inconsistent feature sensitivity, difficulty in understanding high-level context, and excessive dependence on data load. This study identified the limitations of these models and demonstrated that addressing these issues (such as through aligned representations) can significantly improve the performance of the models.

---

> ### Author Rebuttal · Authors · 2025-07-30
>
> Thank you for your constructive and pragmatically-oriented review. We are grateful for your recognition that our work proposes "a new method for evaluating Network Foundation Models" with "concrete insights" about model limitations. To address your comments regarding the applicability and necessity of our proposed framework and the broader impact of our research, we will revise the paper as follows:
>
> 1. Address the applicability of the framework beyond NFMs:
> We will mention that our intrinsic evaluation framework can theoretically evaluate any model that learns latent space representations, but that it is most valuable for models that involve task-agnostic pre-training, as is the case for NFMs. As such, the framework provides particular insight for (i) foundation models that learn general network representations before task-specific fine-tuning, (ii) self-supervised models that learn from unlabeled network data, and (iii) transfer learning approaches where pre-trained representations are adapted to new domains. We will also discuss that while traditional approaches (e.g., neural networks trained using supervised learning techniques) can be evaluated using our metric alignment component, they lack the explicitly extracted latent representations that make geometric analysis and causal sensitivity testing meaningful.
>
> 2. Comment on the necessity of our proposed framework:
> To address your question about framework necessity that points to fundamental limitations in current approaches to network model evaluation, we will mention that since traditional evaluation efforts rely exclusively on downstream task performance, they suffer from critical shortcomings such as (i) performance scores provide no insight into what the models actually learn about network conditions or protocols, (ii) high performance for one task does nor guarantee generalizability to new network domains, (iii) models can achieve good performance through dataset-specific shortcuts rather than genuine network understanding, and (iv) practitioners have no guidance for choosing models or identifying failure modes.
> Our intrinsic evaluation framework addresses these shortcomings by revealing what the models actually learn about network traffic, independent of specific tasks. The practical value of our proposed framework is demonstrated by our finding that addressing geometric deficiencies (anisotropy) leads to measurable performance improvements (+0.001 to +0.064 F1), showing that intrinsic properties directly impact practical utility.
>
> 3. Expand on the significance of NFM and the impact of our research:
> We will mention in Section 1 that network foundation models represent a paradigm shift in network analysis, moving from task-specific models to general-purpose pre-trained representations. Significant aspects of this shift include (i) generalizability: NFMs learn representations that transfer across multiple network analysis tasks (e.g., classification, anomaly detection, traffic analysis); (ii) scale: NFMs process raw packet data and require no domain expert-based feature engineering; and (iii) impact: NFMs are increasingly deployed in production network systems for security, optimization, and monitoring. However, unlike domains such as computer vision and NLP where frameworks for evaluating foundation models already exist, the networking domain lacks systematic approaches for understanding what these NFMs learn about network conditions, protocols or traffic patterns.
> We will discuss in Section 5 that our framework's necessity is validated through several converging lines of evidence, including (i) cross-architecture consistency: geometric patterns remain stable across different model architectures, suggesting fundamental properties of network representation learning; (ii) cross-dataset robustness: our findings replicate across 5 diverse datasets spanning synthetic traces (mobile traffic, intrusion detection) and real-world traces; and (iii) practical relevance: geometric improvements translate to measurable performance gains, demonstrating actionable insights. In short, our proposed framework provides the network community with systematic tools for comparing different NFM architectures objectively, identifying model limitations before deployment, guiding architectural improvements based on intrinsic properties, and understanding failure modes that downstream metrics miss.

---

### Official Review · Reviewer_rL9D · 2025-07-01

**Rating:** 5
**Confidence:** 3

**Summary:**

This paper proposes a systematic intrinsic evaluation framework for Network Foundation Models (NFMs). It analyzes traditional evaluation frameworks and points out their inherent instability issues. This work breaks through the conventional single-dimensional evaluation paradigm that solely relies on downstream task performance, shifting the research focus from downstream task performance to intrinsic representation properties. By devising a three-dimensional methodology encompassing embedding geometry analysis (quantifying spatial utilization and anisotropy), metric alignment evaluation (verifying the capability to capture domain-expert features), and causal sensitivity testing (analyzing protocol perturbations and context perturbations), we conduct in-depth diagnostics on four SOTA models (YaTC, ET-BERT, netFound, NetMamba) across five representative heterogeneous network datasets.

**Dataset Code Accessibility:**

Yes

**Ethical Comments:**

No major ethical concerns are found in this work. The datasets used are all public, the implementation promotes open science, and there is no indication of privacy violations or misuse.

**Ethical Considerations:**

No, there are no or only very minor ethics concerns

**Final Justification:**

The authors' responses have addressed my concerns, and I will raise the score.

**Limitations Weaknesses:**

1. Although the paper conducted experiments on multiple models across different datasets, biases still persist in the experimental results. Specifically, for embedding geometry analysis, only cosine similarity was employed to measure anisotropy, which may lead to misleading conclusions when using pre-training based on cosine loss.
2. The authors conducted causal sensitivity tests but failed to account for confounding variables. The feature correlation analysis in the paper indicates that ET-BERT exhibits low alignment ; however, there was no in-depth exploration into whether this stems from input modalities rather than inherent model flaws.
3. The paper proposes in its abstract that addressing anisotropy can enhance model performance; however, the experiments only validate this through fine-tuning with a linear classifier, without comparing the effects of end-to-end fine-tuning, which may overestimate the performance gains. The linear probing F1 scores attributed to high-level context disentanglement are ascribed to model capability, yet potential issues of dataset specificity and variability are not ruled out.

**Strengths Contributions:**

1.Taking intrinsic representation properties as the entry point, this paper proposes a relatively systematic evaluation framework, filling the research gap in NFMs where only downstream task performance analysis had been the focus. The framework comprises three complementary techniques: embedding geometry analysis, metric alignment evaluation, and causal sensitivity testing. This multi-dimensional approach is not only innovative but also provides reusable evaluation tools, significantly advancing the research on the interpretability of NFMs.
2. The authors obtained a variety of conclusions through experiments conducted on four SOTA models. All NFMs exhibited high anisotropy and demonstrated an excessive reliance on payloads, thereby exposing robustness flaws. Causal sensitivity tests revealed that while NFMs can implicitly capture high-level contexts, they lack sufficient disentanglement capabilities. These conclusions not only uncover the current limitations of NFMs but also validate, through the proposed framework, the potential for improvement.

---

> ### Author Rebuttal · Authors · 2025-07-31
>
> Thank you for your constructive review. We appreciate your recognition of our systematic evaluation that is "not only innovative but also provides reusable evaluation tools." Your technical concerns deal with important evaluation methodology considerations, and we respond to each concern by highlighting our existing comprehensive analysis and targeted new validation and detailing how we will address it and improve the overall presentation.
>
> 1. Concerns regarding cosine similarity
>
> First, we acknowledge your concern regarding reliance on cosine similarity, but would like to mention that this concern is explicitly noted as one of the limitations of our proposed methodology in Section 5. Second, in view of this concern, we explored an alternative metric to characterize the geometric space and reported in Table 9 in Appendix F the results of the manifold dimensionality analysis. These results provide an orthogonal view on the models' embedding geometry, which can be used for comparison in case of models trained using cosine loss.
>
> 2. ET-BERT input modality analysis
>
> Your observation about ET-BERT's low alignment potentially stemming from input modalities rather than model limitations aligns with aspects of our analysis (see Table 5, where ET-BERT shows absolute payload dependency, and Table 9 in Appendix F, where ET-BERT demonstrates very high manifold ID). However, due to the data representation challenges for network traffic (see Appendix A for the details), the choice of input preprocessing is intrinsically entangled with the model architecture and represents the explicit decision of the model's original paper, which requires us to evaluate these parts in combination. We highlight the potential issues with input modalities in Appendices B and F, but we will add further clarification about this in the text of the paper.
>
> 3. a) Linear probing vs end-to-end fine-tuning
>
> We agree that linear probing may indeed provide more optimistic estimates of improvement than end-to-end fine-tuning, where geometric corrections could be overwhelmed by parameter updates throughout the pipeline. However, given our paper’s core argument, we recognize that our claim in the Abstract about "addressing anisotropy can enhance model performance" may indeed be misleading. We will clarify this issue when revising both the paper’s abstract and main text. In particular, please note that the mentioned linear probing experiments serve a very specific and limited purpose: demonstrating the causal effect of anisotropic geometric representations and validating that decorrelation can address these deficiencies. This experiment is not intended as a comprehensive assessment of a model’s ability to perform well on different downstream tasks. This approach aligns with established practices in foundation model evaluation (e.g., CLIP, BERT probing studies) where linear evaluation reveals intrinsic representational quality. Our paper's central contribution is arguing for decoupling NFM assessment from downstream task performance. This decoupling is critical because downstream task evaluation conflates multiple factors (input modalities, pretraining architecture/datasets, fine-tuning architecture/datasets) and obscures what models actually learn about network conditions, protocols, and traffic patterns. The linear probing results demonstrate that geometric properties have measurable effects, but our framework's value lies in revealing intrinsic representational quality independent of task-specific optimization.
>
> 3. b) Cross-dataset generalization validation
>
> The reported results provide evidence that our methodology is largely insensitive to dataset specificity and variability. Our geometric properties remain consistent across diverse datasets drawn from controlled and real-world environments spanning different network types (mobile traffic, intrusion detection, real-world ISP traces). In addition to Table 2, we provide further analysis of linear probing on datasets in Appendix D and zero-shot clusterization in Appendix G, both demonstrating consistent architectural patterns across domains.

---

### Official Review · Reviewer_4byp · 2025-07-03

**Rating:** 5
**Confidence:** 2

**Summary:**

The submission explores the internal representations of Network Foundation Models (NFMs), focusing on their hidden knowledge rather than downstream performance. The authors introduce a three-part evaluation scheme composed of geometry analysis (measuring representation space utilization), metric alignment (measuring similarity to domain-expert features), and causal sensitivity testing (measuring how representations change with protocol and context feature perturbations). Studying four SOTA NFMs across five datasets, the authors find consistent weaknesses like high anisotropy, payload dependence, and poor abstraction of higher-level context. Addressing these issues leads to measurable performance improvements. The work advocates for moving beyond downstream task-based evaluation to motivate a better understanding of NFM architectures, representations, and performance.

**Additional Feedback:**

Although I’m not deeply familiar with networking models, I found the approach in this paper compelling and believe it makes a meaningful contribution to advancing the field. Its focus on intrinsic model representations is both timely and important, especially for developing more generalizable and robust systems.

**Dataset Code Accessibility:**

Yes

**Dataset Code Comments:**

Fully reproducible code is provided and anonymized for the submission.

**Ethical Comments:**

No significant ethical concerns in the submission.

**Ethical Considerations:**

No, there are no or only very minor ethics concerns

**Final Justification:**

I thank the authors for their response to address my concerns by adding an overview figure and improving the presentation. I am keeping my recommended score as "Accept."

**Limitations Weaknesses:**

The paper feels dense for readers unfamiliar with networking or foundation models. A visual overview of the three-part evaluation framework and/or a brief illustration or flowchart of what an NFM is and how it operates would help ground the reader early on. The paper is table-heavy and only has one figure. My recommendation is to insert an introductory figure to the introduction or at the beginning of section 3 to anchor the readers and improve readability.

**Strengths Contributions:**

The submission presents a strong, novel approach to evaluating NFMs beyond downstream metrics. The authors contribute a principled framework that gives many insights into their internal representations. They conduct rigorous experiments across diverse models and datasets, which showcases its relevance and potential impact on future NFM research. The paper is well-organized, compelling, and the metrics chosen for evaluation are well-justified.

---

> ### Author Rebuttal · Authors · 2025-07-30
>
> Thank you for your constructive review. We appreciate your assessment on "a strong, novel approach" with "rigorous experiments across diverse models and datasets" and agree that including more visuals and improving the presentation will significantly enhance the paper's accessibility for the broader research community. To this end, we will make the following changes:
>
> 1. Including an overview figure:
> Following your suggestions for "a visual overview of the three-part evaluation scheme" and "a brief illustration or flowchart of what an NFM is and how it operates”, we will add a single new figure in Section 1 that (i) provides an overview of the NFM processing pipeline with its pre-training and fine-tuning stages, (ii) shows where our three evaluation components (geometry analysis, metric alignment, causal sensitivity) are situated within this pipeline, and (iii) and illustrates the key difference between our newly-proposed intrinsic evaluation framework and traditional frameworks that focus on downstream task evaluation.
>
> 2. Improving the presentation:
> In response to your comment that "the paper feels dense for readers unfamiliar with networking or foundation models”, we will revise the presentation by (i) adding more background context on networking fundamentals and NFM concepts (Section 1), (ii) enhancing the related work discussion to better position our contributions for readers from different AI domains (Section 2), (iii) providing a better motivation for why each evaluation component matters for understanding network models (Section 3), and (iv) including clearer explanations of technical networking terminology (throughout the paper).

---

> > ### Comment · Reviewer_4byp · 2025-08-05
> >
> > Thank you very much for your detailed response. This addresses my concerns, and I am keeping my recommendation as "Accept."

---

### Note · Authors · 2025-08-13

We thank the reviewers who actively participated and provided positive feedback during the rebuttal period. Below we summarize our responses:

Reviewer 4byp
We will follow the reviewer’s suggestions to (i) include an overview figure that illustrates the basic NFM processing pipeline and highlights the components of our proposed three-part evaluation, and (ii) provide more background content on NFMs.

Reviewer rL9D
We clarified that the reviewer’s concern regarding the use of cosine similarity is explicitly noted in Section 5 as one of the limitations of our approach and that Appendix F (Table 9) provides information about the use of an alternative metric (i.e., manifold dimension) for characterizing a model’s embedding geometry.
We addressed the reviewer’s concern regarding the ET-BERT input modality analysis by pointing to relevant results in Appendices B and F and clarified that data representation challenges for network traffic (discussed in Appendix A) require us to evaluate input modalities in conjunction with model architecture.
We acknowledged the reviewer’s concern regarding linear probing potentially overestimating performance gains but clarified that the mentioned linear probing experiments serve a very special and limited purpose and are not meant as a comprehensive assessment model’s ability to perform well on different downstream tasks (i.e., end-to-end fine-tuning).

Reviewer fUnR
As suggested, we performed additional experiments and analyses to provide further insights into the dimensional utilization patterns. In particular, we computed a number of different metrics related to the variability in inter-dimensional similarity and observed that the obtained results are consistent with MCC findings reported in the paper. In addition, we provided more details about our methodology for feature extraction as part of the metric alignment assessment, and enhanced Figure 1 to improve its readability and clarity.

Reviewers fUnR and jicE
We addressed the reviewers' comments regarding the applicability of the proposed framework beyond NFMs and clarified the paper’s central contribution by highlighting the importance of decoupling NFM assessment from downstream task performance that can conflate various factors and obscure what NFMs actually learn during pre-training.

---

### Decision · Program_Chairs · 2025-09-18

**Decision:**

Accept (poster)

**Comment:**

This paper investigates network foundation models (NFMs) via an intrinsic evaluation framework for the latent representation the models have learned rather than the downstream task performance. The framework include embedding geometry analysis to quantify representation space utilization, metric alignment assessment to measure correspondence with domain-expert features, and causal sensitivity testing to analyze protocol and context dependencies.

This work is well appreciated for its new perspective to evaluate NFMs through the lens of internal representations beyond downtream task performance. The evaluation is comprehensive which considers four state-of-the-art NFMs, three evaluation tasks, and five datasets.

Reviewers raised several conconerns about this work: (1) its presentation of background knowledge on networking and NFMs, figures, and experimental results; (2) limitation of linear probing and overclaim about anisotropy; and (3) generalizability to other models/datasets/tasks. There are also other concerns regarding additional analysis or clarity in experimental analysis, which have mostly been discussed in the main content or appendix. Reviewers' concerns are well addressed in rebuttal and/or promised to be explicitly discussed in the revised version.